# Graphene Nanoplatelet-Reinforced Poly(vinylidene fluoride)/High Density Polyethylene Blend-Based Nanocomposites with Enhanced Thermal and Electrical Properties

**DOI:** 10.3390/nano9030361

**Published:** 2019-03-04

**Authors:** Kartik Behera, Mithilesh Yadav, Fang-Chyou Chiu, Kyong Yop Rhee

**Affiliations:** 1Department of Chemical and Materials Engineering, Chang Gung University, Taoyuan 333, Taiwan; b.kartik1991@gmail.com (K.B.); dryadavin@gmail.com (M.Y.); 2Department of General Dentistry, Chang Gung Memorial Hospital, Taoyuan 333, Taiwan; 3Department of Mechanical Engineering, College of Engineering, Kyung Hee University, Yongin 446-701, Korea

**Keywords:** PVDF, HDPE, graphene nanoplatelet, nanocomposites, electrical properties, thermal properties

## Abstract

In this study, a graphene nanoplatelet (GNP) was used as a reinforcing filler to prepare poly(vinylidene fluoride) (PVDF)/high density polyethylene (HDPE) blend-based nanocomposites through a melt mixing method. Scanning electron microscopy confirmed that the GNP was mainly distributed within the PVDF matrix phase. X-ray diffraction analysis showed that PVDF and HDPE retained their crystal structure in the blend and composites. Thermogravimetric analysis showed that the addition of GNP enhanced the thermal stability of the blend, which was more evident in a nitrogen environment than in an air environment. Differential scanning calorimetry results showed that GNP facilitated the nucleation of PVDF and HDPE in the composites upon crystallization. The activation energy for non-isothermal crystallization of PVDF increased with increasing GNP loading in the composites. The Avrami *n* values ranged from 1.9–3.8 for isothermal crystallization of PVDF in different samples. The Young’s and flexural moduli of the blend improved by more than 20% at 2 phr GNP loading in the composites. The measured rheological properties confirmed the formation of a pseudo-network structure of GNP-PVDF in the composites. The electrical resistivity of the blend reduced by three orders at a 3-phr GNP loading. The PVDF/HDPE blend and composites showed interesting application prospects for electromechanical devices and capacitors.

## 1. Introduction

In the past two decades, special attention has been paid to polymeric blend-based nanocomposites, because these systems benefit from the advantages of both blends and nanocomposites [1,2]. Various combinations of polymer matrices and a small amount of nanofillers have been designed to study their potential in improving the chemical and physical properties of neat polymers. The one-dimensional carbon nanotube (CNT) and two-dimensional nanoclays are two of the most studied nanofillers. Reviews on CNT-based polymer nanocomposites have been updated [3,4,5,6]. Additionally, graphene and its derivatives have also been recognized as appropriate nanofillers in fabricating polymer nanocomposites to improve the properties of parent polymers [7,8,9,10]. Two-dimensional graphene or graphene nanoplatelets reveal a 1 TPa Young’s modulus, a 130 GPa ultimate strength, and a high electrical conductivity of 6000 S/cm [11]. Similar to CNT-based nanocomposites, the interfacial adhesion between graphene-polymers and the dispersion of graphene throughout the polymers are the key factors in achieving nanocomposites with an advanced performance [6]. Moreover, dispersion and distribution of the fillers are strongly affected by the melt-mixing methods.

Poly(vinylidene fluoride) (PVDF), a crystalline thermoplastic polymer, shows excellent properties, such as a high mechanical strength, worthy dielectric properties, and remarkable thermal/chemical stability. The disadvantages of PVDF include its high cost and low production volume. The five crystalline polymorphs (α, β, γ, δ, and ε) of PVDF have attracted much academic attention [12,13]. The non-polar α-form with trans-gauche-trans-gauche (TG^+^TG^−^) linkage conformation is the most stable and encountered polymorph. The polar β-form crystal with all-trans (TTTT) zig-zag conformation is the most attractive polymorph because of its unique piezo- and pyroelectric properties, which allow PVDF applications in sensor and actuators. PVDF-based blends and nanocomposites have been investigated for extending the versatility of PVDF [14,15,16,17,18,19,20]. Mago et al. [21] reported that the crystal size of PVDF decreases with increasing CNTs content, whereas its rate of crystallization increases with increasing β-form crystallization. Almasri et al. [20] investigated PVDF/double-walled carbon nanotube (DWNT) nanocomposite films fabricated by the solvent casting method. They found that the storage modulus of PVDF evidently increased by 48% below *T_g_* and increased by up to 85% above it, and the percolation threshold at the addition of 0.23 vol.% DWNTs achieved electrical conductivity. Martins et al. [22] demonstrated that the electrical and rheological percolation threshold was achieved at 1.2 and 0.9 wt.% MWCNT loading, respectively. They showed that a 0.5 wt.% MWCNT nanocomposite revealed a uniform dispersion throughout the PVDF matrix, whereas a percolated network started to form at 1 wt.%. Wang et al. [15] used quaternary phosphorus salt to physically modify graphene and achieved an excellent dispersion of graphene within the PVDF matrix. Moreover, the electrical percolation threshold was achieved at 0.662 wt.%. The nanocomposite material films displayed tremendous electric and dielectric properties, and the salt modifier induced β- and γ-form crystals. Blend-based nanocomposites have received academic and industrial interest because of their potential to exhibit superior properties. Polyethylene (PE) is an important thermoplastic with versatile properties, which has been widely used in the agricultural, automotive, and packaging industry due to its low cost, easy processability, good mechanical/thermal properties, insulation capability, resistance to chemical solvents, and biological attack [23,24,25,26]. Its use has repetitively grown in the plastic industry. However, certain problems exist, caused by the inherent insulation of polymers, thus restricting all application fields, such as the material surface trend for the easy accumulation of charges trapping dust and the deterioration of product performance, possibly resulting in an explosion. Only a few studies have been reported on PVDF/PE blend-based nanocomposites [27,28,29]. Blending of PVDF with a suitable polymer such as PE can be an effective strategy to overcome its drawbacks. The blend of PVDF/HDPE and their conductive composites is especially cost effective and used in micro-electromechanical devices and high-charge storage capacitors. Chiu [30] used CNT, GNP, and organo-montmorillonite (Cloisite 15A) as reinforcing fillers to prepare PVDF/polycarbonate (PC) blend-based nanocomposites. He found that fillers were selectively located in the minor PC phase. The localization of some parts of the Cloisite 15A filler at the interface of the PVDF/PC blend facilitated the crystallization of PVDF and further induced the formation of β-form PVDF. Moreover, Chiu et al. [31] compared the thermal, mechanical, and electrical properties of PVDF/GNP nanocomposites and PVDF/PMMA/GNP blend-nanocomposites. They reported that GNP had a higher nucleation effect on crystallization of the PVDF in ternary composites compared with binary composites. The electrical percolation threshold was achieved at 1–2 phr GNP loading for the two composite systems, whereas ternary composites showed a lower electrical resistivity at identical GNP loadings. Rafei et al. [27] examined the morphology, rheological properties, and electrical conductivity of PVDF/PE/GNP ternary nanocomposites. They reported that double percolation was predicted for the PVDF/PE blend containing 0.9 wt.% GNP and was confirmed by direct electron microscopy and conductivity analysis.

From academic and industrial viewpoints, the influences of the incorporation of GNP on the various properties of PVDF should be thoroughly investigated. GNP-loaded PVDF/HDPE blend-based nanocomposites have been less reported. In the current study, commercialized GNP served as the nanofiller in preparing PVDF/HDPE blend-based nanocomposites through a melt-mixing method. It aimed to investigate the morphology and the resulting physical properties of the prepared samples. The thermal properties (including crystallization/melting behavior and thermal stability) of neat components, a blend, and composites were compared and reported. The mechanical and rheological properties and electrical resistivity of the samples were also characterized.

## 2. Materials and Methods

All materials used in this study are commercially available. PVDF with an average molecular weight of 1.8 × 10^5^ g/mol in pellet form (Kynar 710, Arkema, Colombes, France) was used as the major component in the blend/composites. Commercial grade HDPE (Taisox 8050, MFR-6.0 g/10 min, density of 0.96 g/cm^3^) was supplied by Formosa Plastic Corporation, Taoyuan, Taiwan. The graphene nanoplatelet (xGNP M-15, denoted as GNP), obtained from XG Sciences, Inc. (Lansing, MI, USA), was used as the nanofiller for composites fabrication. The components were pre-dried in a vacuum oven at 70 °C for 24 h to eliminate the absorbed moisture and then dry-mixed at given ratios before being fed into the mixer. All blend/composites samples were mixed using a Thermo Haake PolyDrive mixer (R600, Karlsruhe, Germany) at 190 °C for 10 min with a rotor speed of 60 rpm. The PVDF/HDPE blend was prepared with the weight ratio of 70:30. GNP was loaded at concentrations of 0.5–3 phr into the blend to prepare the composites. Neat PVDF and HDPE were also melt-treated for the purpose of comparison. The sample designation of F7E3 represents the PVDF/HDPE-70/30 blend. The codes of F7E3-# represent the # phr (1 phr = 0.99 wt.%, 2 phr = 1.96 wt.%, and 3 phr = 2.91 wt.%) of GNP loaded in the composites. To attain different shapes of specimens for further characterizations, the melt-mixed samples were consequently compression-molded at 190 °C for 3 min.

The fractured surface morphology of the blend and composites was examined using a field emission scanning electron microscope FESEM, Jeol JSM-7500F (Akishima, Tokyo, Japan). The cryo-fractured (in liquid nitrogen for 3 min) specimens were sputter-coated with gold prior to observation. Thermogravimetric analysis (TGA) was conducted by using a TA Q50 analyzer (TA instrument, New Castle, DE, USA) under air and nitrogen environments at a 10 °C/min heating rate. The crystallization kinetics and the resulting melting behavior of the samples were studied using a differential scanning calorimeter (DSC, TA instrument, New Castle, DE, USA) TA Q10. For non-isothermal crystallization experiments, the samples were first melted at 210 °C for 2 min to erase the thermal history. The samples were then cooled at various rates to 20 °C. The cooled samples were subsequently heated to 210 °C at 10 °C/min to evaluate their melting behavior. For isothermal crystallization kinetics study, samples were rapidly cooled at a rate of 100 °C/min (from 210 °C) to the pre-set crystallization temperatures (*T*_c_), and the crystallized samples were then heated at 10 °C/min to evaluate the melting behavior. The crystal structures of PVDF and HDPE in the blend/composites were investigated using an X-ray diffractometer (AXS, D2, Bruker, Karlsruhe, Germany) operating at 40 kV and 40 mA. The X-ray source was CuK_α_ radiation with a wavelength of 1.54 Å.

Fourier-transform infrared spectroscopy FTIR spectra of the samples were recorded using a BRUKER, TENSOR 27 IR spectrometer (Karlsruhe, Germany). The samples were analyzed within the range of 700–2500 cm^−1^. A Raman spectrometer (UniDRON, C. L. Technology Co., Ltd., New Taipei City, Taiwan) with a 532 nm laser excitation wavelength was used. All samples were compression molded into films at 200 °C and 100 bar. The samples were cooled to room temperature in the mold under pressure. The rheological properties of the samples were analyzed at 190 °C by using an Anton Paar Physica rheometer (MCR 101, Graz, Austria) in an oscillating mode. A parallel plate geometry with a 25 mm diameter and 1 mm gap was used, and the strain amplitude was set at 1%. The tensile properties of the compression-molded specimens (according to ASTM D638) were measured at a crosshead speed of 5 mm/min with a Gotech AI-3000 system (Taichung, Taiwan). The flexural modulus of the samples (according to ASTM D790) was measured using the same Gotech AI-3000 system at a cross-head speed of 5 mm/min. Izod impact tests (according to ASTM D256) were performed using a CEAST impact tester (Taichung, Taiwan). The reported data was expressed as the average of at least five specimens of the same formulation. The volume electrical resistivity of each specimen was measured using commercial resistivity meters (MCP-HT450 and MCP-T600, Mitsubishi Chemical Co., Yamato, Japan) at room temperature. The resistivity data was expressed as the average value of four repeated measurements at different positions.

## 3. Results

### 3.1. Phase Morphology and Selective Localization of GNP

The cryo-fractured surfaces of the prepared blend and blend-based composites were investigated using FESEM, as shown in Figure 1. Figure 1a depicts the F7E3 (PVDF70/HDPE30) blend, which evidently exhibited a bi-phasic (matrix-dispersed domain) morphology, indicating its immiscible character. The dispersed HDPE phase (minor component) exhibited a distribution of spherical domain size within the PVDF matrix phase, and the domain ranged from 8–30 µm. The size of domains was measured by using eight domains. The displayed piercing boundary clearly indicated no specific interaction between the PVDF and HDPE phases. Figure 1b–d depict the images of GNP-added representative composites. The domain size of HDPE decreased after the addition of GNP, and the size further decreased with increasing GNP loading. The added-GNP might have prevented the aggregation of HDPE domains forming large HDPE domains. Notably, the HDPE domains had a continuous-like morphology in the composites, which was more evident at high GNP loading. The GNP (arrowed) was mainly located within the PVDF matrix, which suggested that the interaction between PVDF and GNP was greater than that of HDPE and GNP. Thus, GNP thermodynamically prefers the PVDF phase. The inset Figure 1b clearly reveals the dispersion of GNP in the PVDF matrix. Additionally, the localization of GNP in the PVDF matrix caused the increased viscosity of the composites at the mixing temperature (see rheological data). According to the morphological observations, GNP was well-dispersed in the PVDF matrix phase, thereby confirming the achievement of blend-based nanocomposites. The thermal and mechanical properties of the F7E3 blend were improved after forming the nanocomposites. The results are discussed in the following sections.

The selective localization of filler in the immiscible polymer blend was executed by kinetic and thermodynamic factors. Sumita et al. [32] reported that through the wetting coefficient (*ω*) from Young’s equation, the GNP localization could be predicted by the following equation in equilibrium:(1)$$\omega =\frac{\gamma_{\mathrm{HDPE}-\mathrm{GNP}\;}-\;\gamma_{\mathrm{PVDF}-\mathrm{GNP}}}{\gamma_{\mathrm{PVDF}-\mathrm{HDPE}}}$$where *γ*_PVDF-GNP_, *γ*_HDPE-GNP_, and *γ*_PVDF-HDPE_ are the interfacial tension between PVDF and GNP, HDPE and GNP, and PVDF and HDPE phases, respectively. If *ω* > 1, the GNP is localized in the HDPE phase. If −1 < *ω* < 1, GNP will be located at the interface of the two polymer phases. If *ω* < −1, GNP will preferentially reside in the PVDF phase. The values of interfacial tension between different polymers can be evaluated from their surface energies based on the Harmonic-mean equation and the Geometric-mean equation:(2)$$\gamma_{12}=\gamma_{1}+\gamma_{2}-4\left(\frac{\gamma_{1}^{d}\gamma_{2}^{d}}{\gamma_{1}^{d}+\gamma_{2}^{d}}+\frac{\gamma_{1}^{p}\gamma_{2}^{p}}{\gamma_{1}^{p}+\gamma_{2}^{p}}\right)$$
(3)$$\gamma_{12}=\gamma_{1}+\gamma_{2}-2\left(\sqrt{\gamma_{1}^{d}\gamma_{2}^{d}}+\sqrt{\gamma_{1}^{p}\gamma_{2}^{p}}\right)$$
where, *γ^d^* and *γ^p^* stand for the dispersive and polar parts of surface tension, respectively. The provided surface tension values of PVDF, HDPE, and GNP in Table 1 were followed by reported data [27,33,34]. The interfacial tension at 190 °C between the components was calculated using Equations (2) and (3) (see Table 2). According to the theoretical prediction by Equation (1), the *ω* values calculated by the harmonic-mean and geometric-mean equations were −1.37 and −1.17, respectively (Table 2), confirming the dispersion of GNP within the PVDF matrix during melt-mixing. These theoretical results agree with the SEM results, as discussed in the above section.

### 3.2. Thermal Stability

The TGA curves of the selected samples scanned under N_2_ and air environments are shown in Figure 2. Figure 2a shows that neat HDPE displayed an evidently lower degradation temperature than PVDF under an N_2_ environment. The temperature at 10% weight loss (*T*_d10_) was 451 and 422 °C for PVDF and HDPE, respectively. The degradation marginally shifted to a higher temperature for the composites compared with that of the blend, and a higher GNP loading led to a higher degradation temperature. The thermal stability increased for the GNP-added composites, which was attributed to the high aspect ratio of GNP that served as a barrier and then prevented the emission of gaseous molecules during the thermal degradation. In addition, the radical scavenging function of GNP could inhibit the degradation process of the organic polymers. The GNP enhanced the thermal stability in some reported nanocomposite system [35]. The decomposition curves of the samples scanned under an air environment are compared in Figure 2b. The samples scanned in an N_2_ environment exhibited higher degradation temperatures compared with those scanned in air because of the auto-oxidation reaction (due to the presence of oxygen). The neat PVDF again exhibited a higher degradation temperature than that of HDPE. The *T*_d10_ was 446 °C for PVDF and 376 °C for HDPE under an air environment. As anticipated, the F7E3 blend curve lied between those of PVDF and HDPE. After the formation of the blend and composites, two-step degradation corresponding to the individual degradation of PVDF and HDPE was observed. The composites displayed an improved degradation behavior compared to that of the F7E3 blend. The increase in thermal stability was demonstrated more evidently in the PVDF phase, because GNP was located in the PVDF phase (see above FESEM images). The temperatures at 10 and 50 wt.% loss (*T*_d10_ and *T*_d50_) of the samples scanned under both air and N_2_ environments are listed in Table 3. For the blend and composites, *T*_d10_ is mainly associated with the degradation of HDPE and *T*_d50_ corresponds to the degradation of PVDF.

### 3.3. Crystallization Kinetics

Figure 3a shows the DSC cooling thermograms (from the melt state) of the prepared samples at 5 °C/min. Neat PVDF and HDPE showed their crystallization peak temperatures (*T*_p_s, temperature at the exotherm minimum) at 138.1 and 118.2 °C, respectively. After blending with each other, the *T*_p_ value of PVDF remained unchanged and the crystallization temperature of HDPE slightly decreased. Two further observations can be made from this figure. First, the addition of GNP into the blend increased the *T*_p_ of individual PVDF and HDPE, and a higher GNP loading increased the *T*_p_ value. This observation suggests that the GNP acted as the nucleation agent for PVDF and HDPE crystallization. The determined *T*_p_ values of different samples are listed in Table 3. The formulation-dependent crystallization behavior of PVDF and HDPE at 5 °C/min-cooling was similarly observed in the samples cooled at 40 °C/min (Figure 3b). Lower *T*_p_ values were displayed for the samples cooled at a faster rate, mainly due to the thermal lag effect. For kinetic analysis, the crystallization of PVDF in the selected samples under different cooling rates was examined, as exhibited in Figure 3c–e. The *T*_p_ value decreased with the increased cooling rate for all samples, as anticipated. The GNP-added composites showed higher *T*_p_ values compared with the blend at each cooling rate. For example, F7E3 showed a *T*_p_ value ca. 141.6 °C at 2.5 °C/min cooling, whereas F7E3-3 had a *T*_p_ value ca. 142.7 °C at the same cooling rate. The activation energy of PVDF was studied due to its major component. The activation energy (∆*E*_c_) for the non-isothermal crystallization of PVDF in different samples (PVDF used as a major component in the whole experiment) was calculated using the Kissinger equation [36,37,38] (Equation (4)), as the typical plots shown in Figure 3f.
(4)$$\left(\frac{\beta }{T_{p}{}^{2}}\right)=\mathrm{Const}-\frac{\Delta E_{c}}{RT_{p}}$$
where *T*_p_, *R*, and *β* stand for the crystallization peak temperature, universal gas constant, and cooling rate, respectively. The ∆*E*_c_ values of the representative samples are summarized in Table 3. Neat PVDF showed an absolute ∆*E*_c_ value of 303 kJ/mol, while the F7E3 blend showed ∆*E*_c_ values higher than that of neat PVDF. The composites showed that absolute ∆*E*_c_ increased with increasing GNP content in the blend. This result suggests that although GNP accelerated the nucleation of PVDF, it retarded the subsequent crystal growth of PVDF during the non-isothermal crystallization process. The isothermal crystallization kinetics of the selected samples was also analyzed. Figure 4a–d depict the DSC isothermal crystallization curves of the representative samples at different *T*_c_s. According to the nucleation-controlled crystal growth theory, the shift in crystallization exothermic peak time from a lower to higher value with increasing *T*_c_ for individual samples was shown. The plots in Figure 4e show the reciprocal of the crystallization peak time (*t*_p_) as a function of *T*_c_. The reciprocal value (*t*_p_^−1^) was proportional to the overall isothermal crystallization rate. These results specified three pieces of information: First, the crystallization rate decreased with increasing *T*_c_ for the individual sample; second, the F7E3 blend revealed a lower overall crystallization rate than neat PVDF at identical *T*_c_; and third, the composites (i.e., F7E3-1 and F7E3-3) crystallized at a faster rate than the blend at an identical *T*_c_, due to the GNP nucleation effect. The Avrami equation (Equation (5)) [39] was used to evaluate the isothermal crystallization kinetics of PVDF in different samples based on the *X_t_* (crystallinity at time *t*) versus *t* data.
(5)$$X_{t}\;=1-\exp\left(-Kt^{n}\right)$$
where *K* is the crystallization rate constant and *n* is the Avrami exponent, depending on the type of nucleation and crystal growth geometry. Equation (5) can be transformed into the following equation:(6)$$\ln\left[-\ln\left(1-X_{t}\right)\right]=\ln\;K\;+\;n\;\ln\;t\;\;$$

Figure 5a–d show the Avrami plots of the representative samples. The *n* and *K* values were determined from the linear region of the plots (slope and intercept). The *n* value provides qualitative information on the nature of nucleation and crystal growth geometry. The *n* value for PVDF ranged from 1.9–3.8 in different samples and decreased with increasing *T*_c_, as shown in Figure 5. Furthermore, *n* decreased slightly after forming the composites. The decreased *n* values in the composites might indicate that nucleation changed from a thermal to athermal type, which was associated with the nucleation effect of GNP. The growth of PVDF crystals might also have changed from two-dimensional (at high *T*_c_s) to three-dimensional (at low *T*_c_s). The *K* value was noted to decrease with increasing *T*_c_ (not shown for brevity), corresponding to the *t*_p_^−1^ results.

### 3.4. Melting Behavior

Figure 6a illustrates the DSC melting curves (10 °C/min heating) of the 5 °C/min pre-cooled samples. The melting temperatures (*T*_ms_) of neat PVDF and HDPE were ca. 169.2 and 135.0 °C, respectively. The *T*_m_ of individual PVDF and HDPE slightly decreased in the blend. A higher loading of GNP slightly increased the *T*_m_ of PVDF and HDPE. This result suggests that PVDF crystals with a higher stability (more perfect) formed in the composite. The melting behavior of the 40 °C/min pre-cooled samples was also examined, as shown in Figure 6b. The neat PVDF and blend/composites showed multiple peaks from 165–170 °C. The multiple peaks were caused by the melting of original grown and heating-scan annealed α-form crystals (no β-form crystal was detected in the XRD data) [31]. The samples pre-cooled at 5 °C/min did not exhibit melting of the heating-annealed α-form crystals, because the slow cooling rate resulted in the growth of crystals with a sufficient stability. Accordingly, crystal annealing was prevented during heating scans. The *T*_m_ values of PVDF at different samples are summarized in Table 3. Figure 6c–f illustrate the melting curves of the representative samples that were isothermally crystallized at different *T*_c_s. Some features were noticed in these figures. First, the *T*_m_ of PVDF increased with increasing *T*_c_ for the individual samples. Second, neat PVDF, the blend, and F7E3-1 showed similar melting behavior from 168–172 °C. Furthermore, the composites showed lower *T*_m_ values compared with the blend at each *T*_c_. For example, F7E3 showed a *T*_m_ of 170.3 °C at *T*_c_ = 150 °C, whereas F7E3-3 had a *T*_m_ of 169.4 °C at an identical *T*_c_. This is because the PVDF in composites displayed greater super cooling (due to higher *T*_m_ values) compared with in the blend, while they were crystallized at an identical *T*_c_.

### 3.5. Crystal Structure, FTIR, and Raman Spectra

Figure 7 shows the XRD patterns of the 5 °C/min pre-cooled samples. Neat PVDF revealed characteristic diffraction peaks at 2θ values of approximately 17.8° (100), 18.5° (020), 20.1° (110), and 26.8° (021) of the stable α-form [30,31], whereas neat HDPE exhibited characteristic peaks (orthorhombic structure) at 2θ values of approximately 21.6° (110) and 24.0° (200) [25]. The diffraction peaks of individual neat PVDF and HDPE with a lower intensity were observed in the blend and composites. The change in intensity ratio [(mainly between (020) and (110)] of PVDF in the blend/composites is because the growth of the (020) plane was somehow retarded compared to the growth of the (110) plane for PVDF crystals. Additionally, the composites exhibited a strong diffraction peak at ca. 2θ = 26.5, due to the layered structure of GNP. XRD results thus indicated that the crystal structure of PVDF and HDPE remained after forming the blend and composites. The presence of GNP did not affect the crystal structures of PVDF and HDPE.

Figure 7b shows the FTIR spectra of neat PVDF, HDPE, and selected composites. The bands located at 1381 and 1402 cm^−1^ correspond to the deformed vibration of the CH_2_ groups in neat PVDF. The peaks at 1068, 1146, 1179, and 1210 cm^−1^ were related to the CF_2_ stretching mode of PVDF chains [40,41]. The peaks at 762, 795, 839, 872, and 974 cm^−1^ indicated the high content α-form crystal in the neat PVDF sample [41]. F7E3 showed two extra peaks compared with the neat PVDF at the wave numbers of 721 and 1466 cm^−1^, corresponding to the bending and rocking mode of the CH bonds of HDPE chains, respectively [42]. The peaks interrelated to the stretching mode of CH_2_ in both PVDF and HDPE chains overlapped in the F7E3 blend, leading to broadened peaks (as shown in selected area). The intensity of the stretching of CH and CF_2_ bonds of the composites increased compared with the F7E3 blend. The characteristic bands of composites were moderately broadened compared with the blend due to the charge-transfer-type interaction [43] between the electron pairs of fluorine atoms in PVDF and electrons in GNPs.

Figure 7c shows the Raman spectrum of the neat GNP and selected composites. Neat GNP showed a G band at 1569 cm^−1^, which was attributed to the first-order scattering of the E_2*g*_ vibration mode, and the D band at 1336 cm^−1^ arose from the breathing mode of the j-point phonons of A_1*g*_ symmetry [44]. The location of the D band shifted from 1336 to 1362 cm^−1^ while GNP was loaded into the blend, suggesting a significant interaction between PVDF and GNP. Additionally, the G band of GNP shifted from 1569 to 1575 cm^−1^, due to the charge-transfer interaction between the fluorine atoms of PVDF and the aromatic structure of GNP, noted as F–C bonding [42].

### 3.6. Melt Rheology

The rheological property measurements, including complex viscosity (*ղ**) and storage modulus (*G*′), can reveal the processability and change in the internal structure of the neat components after forming the blend and composites. Figure 8a provides a comparison of the *ղ** as a function of sweep frequency (ω) at 190 °C for the selected samples. The neat PVDF and HDPE showed similar Newtonian fluid behavior at the low-frequency region, whereas that in the high-frequency region exhibited non-Newtonian fluid (shear thinning) behavior. The F7E3 blend exhibited non-Newtonian fluid behavior of *ղ** values at all frequencies, and basically in-between that of individual PVDF and HDPE, except for displaying lower values than HDPE at high frequencies. The network structure of PVDF in the blend should have played a role for the observation. The *ղ** values evidently increased at all frequencies after the addition of GNP into the blend, and higher *ղ** values were observed at higher GNP contents. The selective localization of GNP within the PVDF matrix phase along with the formation of the PVDF-HDPE co-continuous morphology should be responsible for this observation. Moreover, *ղ** exhibited a non-Newtonian fluid behavior at all frequencies for the composites. The shear thinning behavior at low frequencies indicated the development of a GNP-in-PVDF pseudo-network structure (liquid-like to solid-like transition) in the composites. Figure 8b shows the *G*′ versus ω plots of the representative samples. *G*′ increased with increasing ω for all samples, and the values increased with the addition of GNP into the blend. The addition of 3 phr GNP resulted in the highest increase in *G*′. Additionally, the smaller slopes of the blend/composite plots compared with those of neat PVDF and HDPE at low frequencies suggest a solid-like behavior, which agrees with the *ղ** observation. The smaller slope values revealed more elastic characteristics of the blend and composites due to the pseudo-network structure (PVDF-GNP) development within the matrix.

### 3.7. Mechanical Properties

The mechanical properties of the prepared samples, including the Young’s modulus (YM), flexural modulus (FM), and impact strength (IS), were measured. The neat PVDF had a YM value (1216 MPa) that was higher than that of the neat HDPE (665 MPa). The F7E3 blend showed a YM value (575 MPa) that was lower than those of its neat components, because of the immiscible character. The YM values and standard deviations (σ) of all samples are summarized in Table 4. The value increased after the GNP was loaded into the blend, and increased up to 2 phr GNP loading. F7E3-3 (654 MPa) showed a value lower than F7E3-2 (697 MPa, 21% increase at 2 phr GNP loading compared with the blend), which could be associated with the agglomeration of GNP at a higher loading. The noticeable improvement in YM after the GNP addition was mainly caused by the intrinsic high strength, high aspect ratio, and adequate interaction between the GNP and PVDF matrix phase [31]. The FM values of the individual samples were also compared and the values are listed in Table 4. PVDF possessed a much higher FM than that of HDPE, and the FM of HDPE significantly increased after blending with PVDF. The addition of GNP caused an evident increase in FM for the composites (24% increase at 2 phr GNP loading compared with the blend). The reasons behind this significant improvement are identical to those of the YM improvement after GNP addition. The notched IS values of the individual sample are listed in Table 4. The neat PVDF possessed an IS of 17.9 J/m, which evidently decreased after the formation of the blend because of the immiscibility. After the addition of 2 phr GNP, IS increased to approximately 1.54 times the original value of the F7E3 blend, and a higher GNP loading slightly reduced the IS value. The GNP was located in the PVDF matrix phase near the interface of PVDF-HDPE, leading to adhesion enhancement of the interface. The possibility of stress transfer between blend components through the potential bridging effect of the GNP at the blend interface caused the enhancement of toughness.

### 3.8. Electrical Resistivity

The uniform distribution of electrically conductive fillers in the polymer matrices offers great potential for fabricating (semi)conductive polymers. The volume electrical resistivity values of the selected samples are summarized in Table 4. Neat PVDF, HDPE, and F7E3 exhibited resistivity values higher than 10^11^ Ω-cm, indicating their electrically insulating characteristic. PVDF shows an electrical resistivity value of 10^11^ Ω-cm, similar to a previously published paper [31]. However, the low loading of GNP (< 3 phr) could evidently reduce the resistivity for the composites. For example, the loading of 0.5 phr GNP decreased the resistivity value from 10^11^ to 10^10^ Ω-cm. The resistivity further decreased (down to 10^8^ Ω-cm) with increasing GNP loading in the composites. The apparent decrease in resistivity was noticed at 1–1.5 phr GNP loading. The observations of a ca. three orders of magnitude drop resulted from the fine dispersion of GNP throughout the PVDF matrix phase. This result is consistent with the above SEM morphological observations.

## 4. Discussion

PVDF/HDPE blend and blend-based nanocomposites with PVDF as the major component were fabricated through the melt-mixing process. SEM results revealed that GNP was mainly located in the PVDF matrix phase. DSC data showed the nucleation effect of GNP on both PVDF and HDPE crystallization. The activation energy for the non-isothermal crystallization of PVDF increased from 303 kJ/mol in the neat state to 374 kJ/mol in the F7E3-3 composite. GNP increased the isothermal crystallization rate of PVDF in the composites. The Avrami *n* values ranged from 1.9–3.8 for PVDF in different samples, and the GNP caused an athermal nucleation mechanism for PVDF crystallization. TGA results confirmed the improvement in thermal stability of the blend after GNP addition under both air and nitrogen environments. XRD results revealed that the presence of GNP did not alter the crystalline polymorph of PVDF and HDPE. An increase in the rheological properties of *ղ** and *G’* was observed with increasing GNP loading. The pseudo-network structure was achieved in the composites at 3-phr loading GNP into the blend. The rigidity of the PVDF/HDPE blend increased with the incorporation of GNP. The toughness of the blend also increased with GNP inclusion, up to a 55% rise after 2 phr loading. The electrical resistivity of the blend dropped by more than three orders of magnitude with a GNP loading of 3 phr. The obtained results indicated that the prepared composites should have great potential applications in micro-electromechanical devices and storage capacitors.

## Figures and Tables

**Figure 1 nanomaterials-09-00361-f001:**
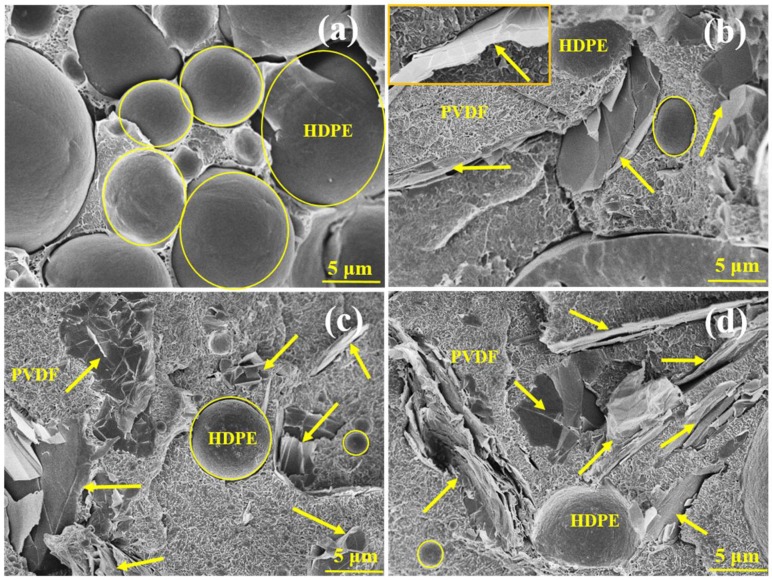
Scanning electron microscope (SEM) images of selected samples: (**a**) F7E3, (**b**) F7E3-1 (higher magnification image included in the inset), (**c**) F7E3-2, and (**d**) F7E3-3.

**Figure 2 nanomaterials-09-00361-f002:**
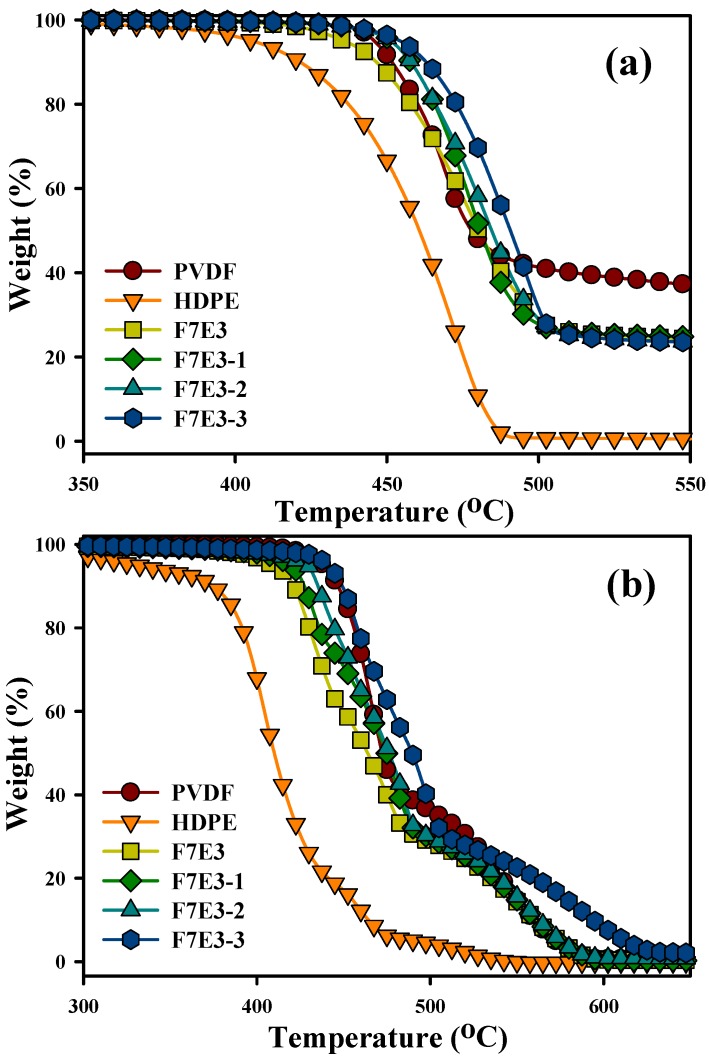
Thermogravimetric analysis (TGA)-scanned curves of selected samples under (**a**) N_2_ environment and (**b**) air environment.

**Figure 3 nanomaterials-09-00361-f003:**
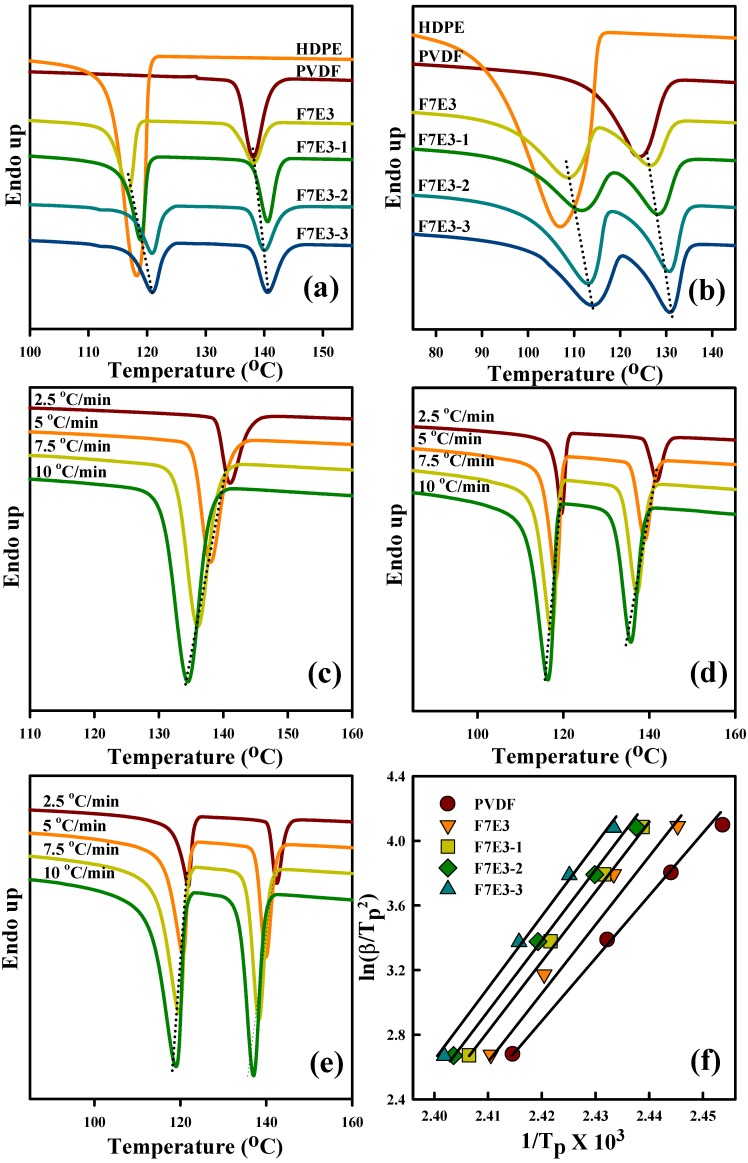
Differential scanning calorimeter (DSC) curves of the selected samples at (**a**) 5 °C/min cooling from the melt, (**b**) 40 °C/min cooling from the melt; DSC cooling curves of (**c**) PVDF, (**d**) F7E3, and (**e**) F7E3-3 at various rates; (**f**) Kissinger plots of selected samples for calculation of crystallization activation energy.

**Figure 4 nanomaterials-09-00361-f004:**
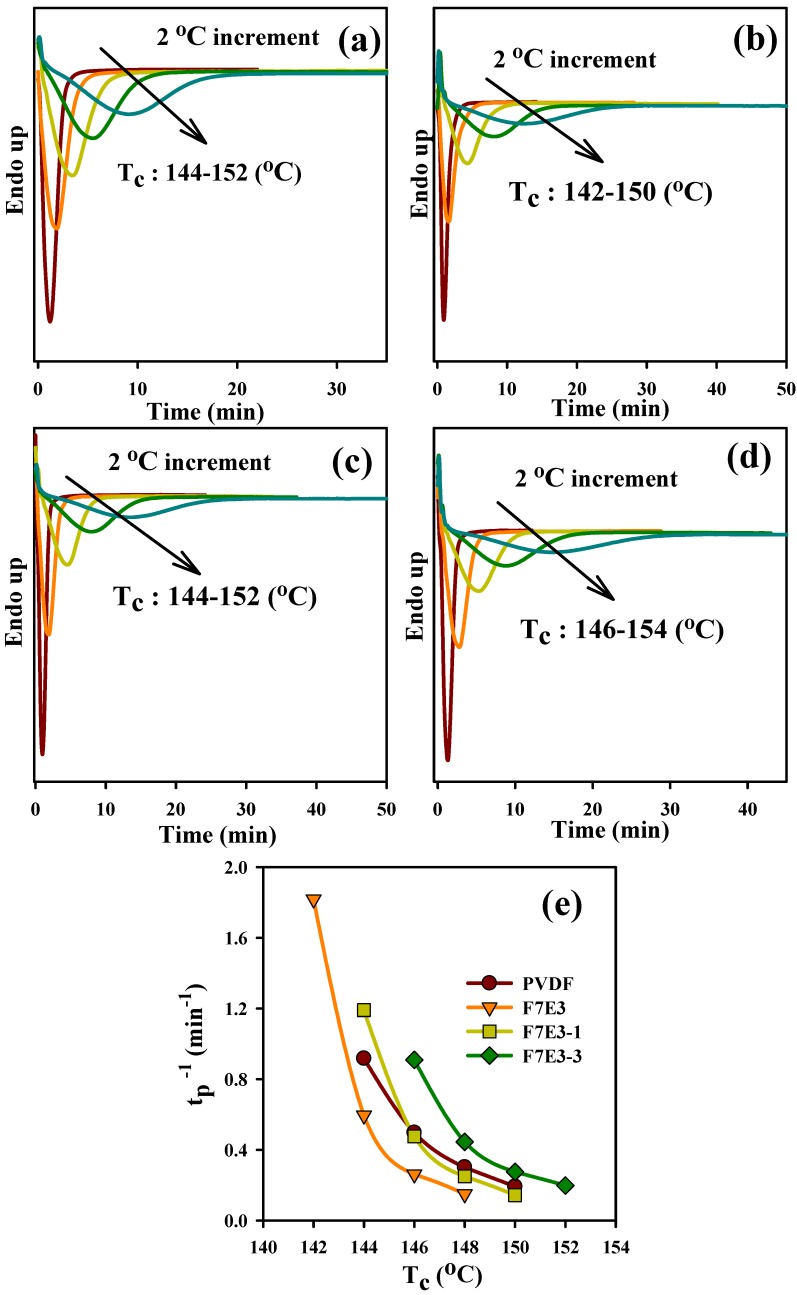
DSC traces of PVDF isothermally crystallized at indicated *T*_c_s in different samples: (**a**) PVDF, (**b**) F7E3, (**c**) F7E3-1, (**d**) F7E3-3, and (**e**) *t*_p_^−1^ vs. *T_c_* of PVDF in representative samples.

**Figure 5 nanomaterials-09-00361-f005:**
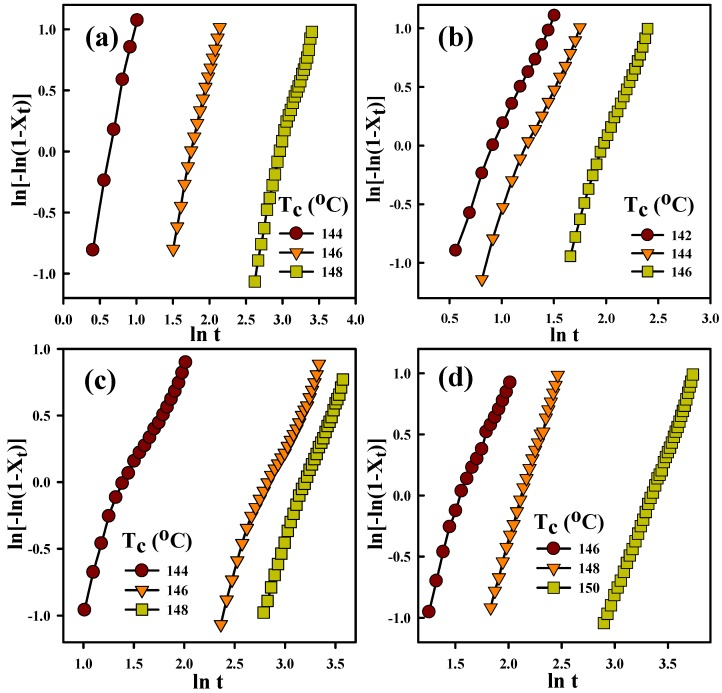
Avrami plots of PVDF isothermally crystallized at different *T*_c_s in different samples: (**a**) PVDF, (**b**) F7E3, (**c**) F7E3-1, and (**d**) F7E3-3.

**Figure 6 nanomaterials-09-00361-f006:**
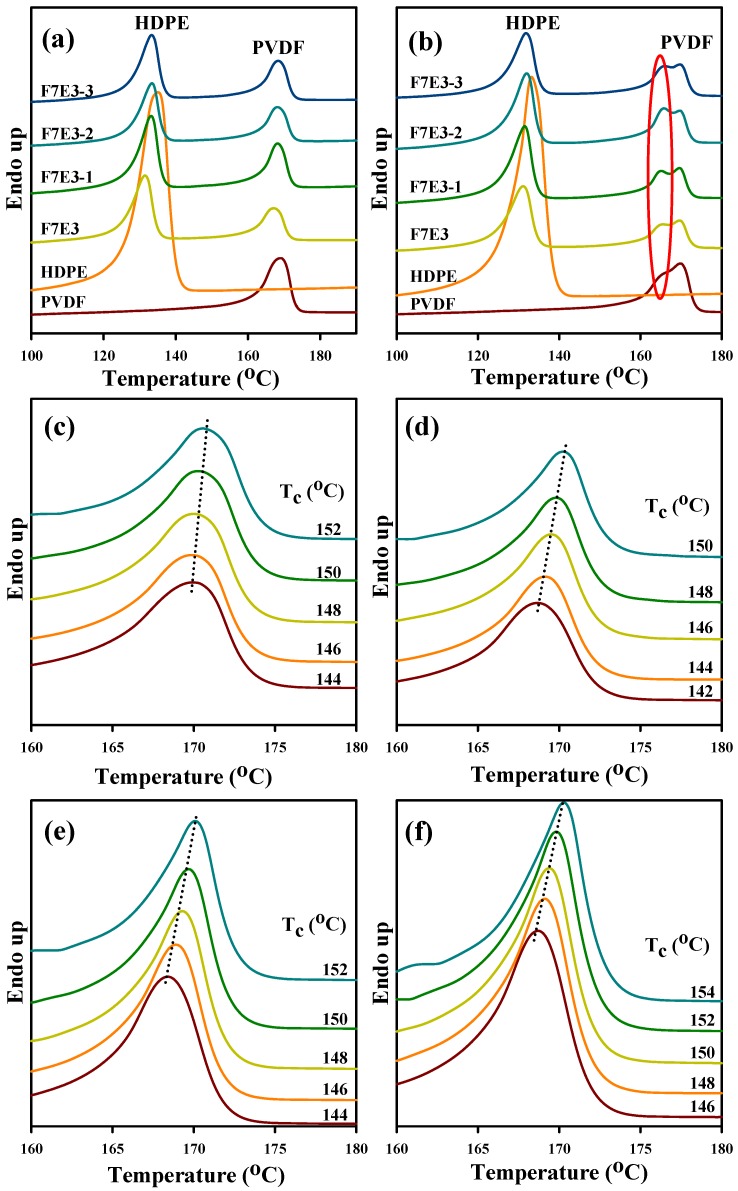
DSC heating curves of different samples: (**a**) 5 °C/min pre-cooled and (**b**) 40 °C/min pre-cooled; DSC melting curves of PVDF isothermally crystallized at indicated *T*_c_s in different samples: (**c**) PVDF, (**d**) F7E3, (**e**) F7E3-1, and (**f**) F7E3-3.

**Figure 7 nanomaterials-09-00361-f007:**
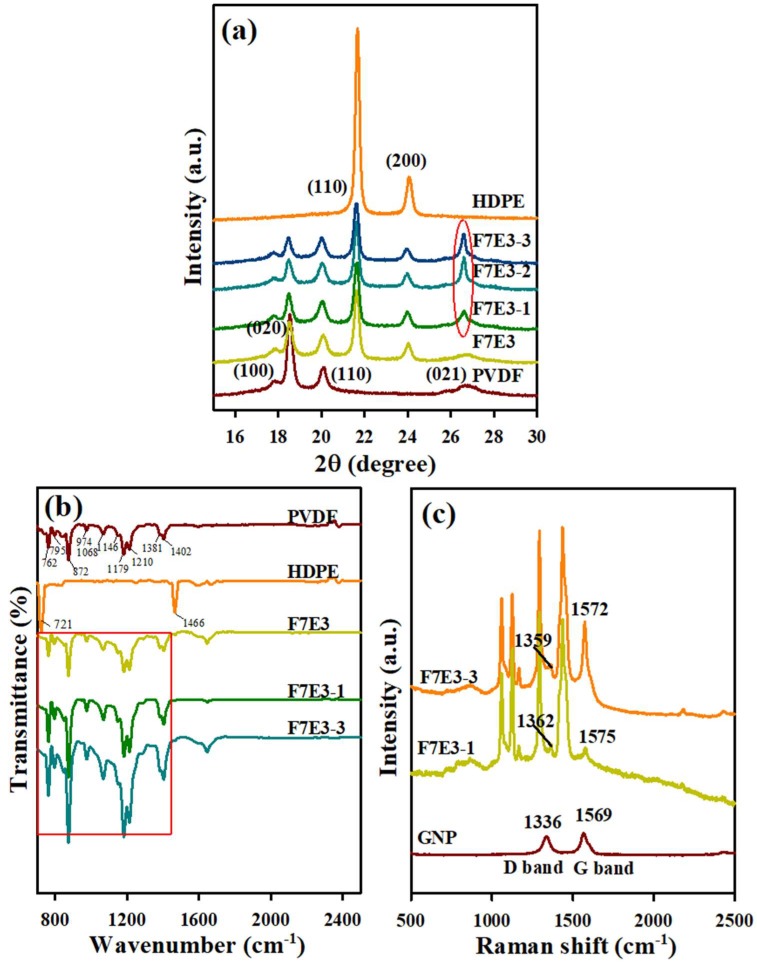
(**a**) X-ray diffraction (XRD) patterns of 5 °C/min-cooled representative samples; (**b**) Fourier-transform infrared spectroscopy (FTIR) spectra of selected samples; and (**c**) Raman spectra of GNP, F7E3-1, and F7E3-3.

**Figure 8 nanomaterials-09-00361-f008:**
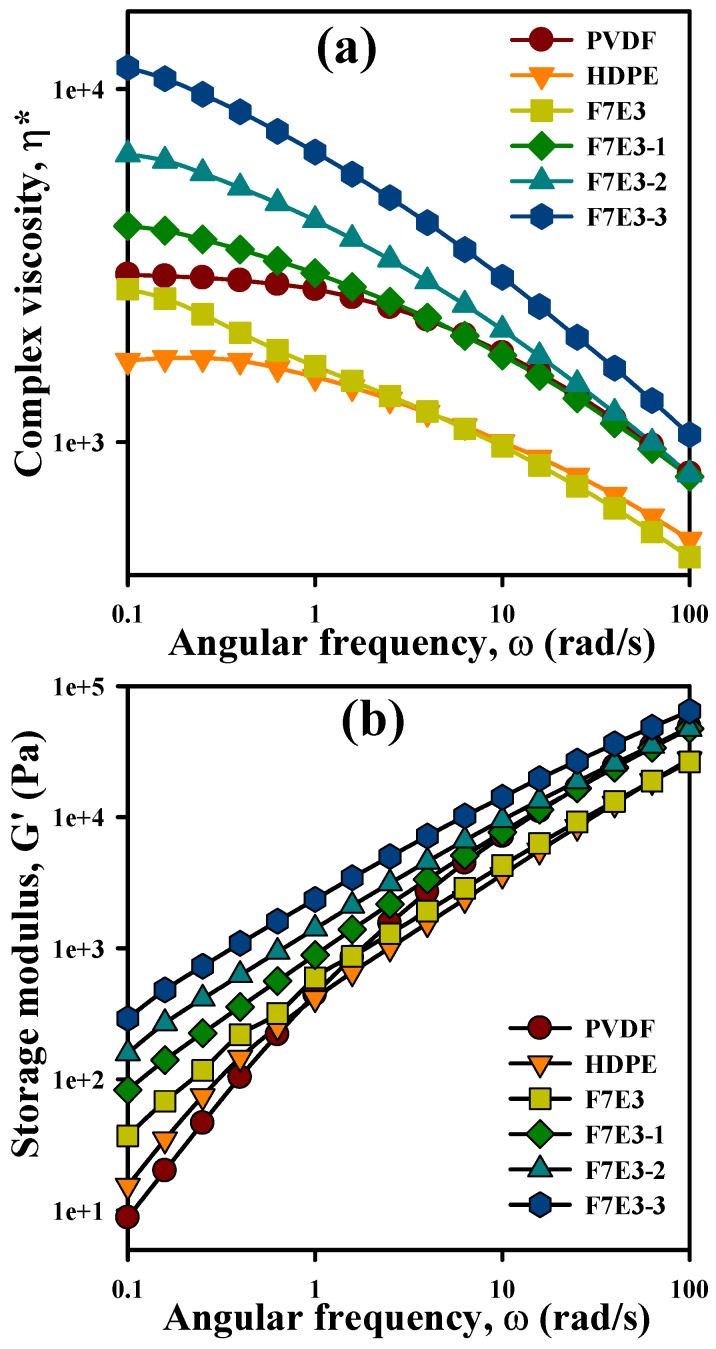
Rheological properties of the selected samples: (**a**) *ղ** vs. ω and (**b**) *G*′ vs. ω.

**Table 1 nanomaterials-09-00361-t001:** Surface tension data of PVDF, HDPE, and GNP at 190 °C.

Samples	*γ* (mN m^−1^)	*γ^d^* (mN m^−1^)	*γ^p^* (mN m^−1^)	References
PVDF	38.0	32.6	5.4	[27]
HDPE	25.9	25.9	0	[33]
GNP	52.6	47.7	4.9	[34]

**Table 2 nanomaterials-09-00361-t002:** Interfacial tension and calculated wetting coefficient of composites at 190 °C.

Samples	*γ* _PVDF-GNP_	*γ* _HDPE-GNP_	*γ* _PVDF-HDPE_	*ω* (GNP)
Harmonic-mean Equation (2)	2.90	11.35	6.17	−1.37
Geometric-mean Equation (3)	1.45	8.20	5.78	−1.17

**Table 3 nanomaterials-09-00361-t003:** Thermogravimetric analysis (TGA) and differential scanning calorimeter (DSC) data of selected samples.

Samples	Properties						
	(*T*_d10_) ^a^ (°C)	(*T*_d50_) ^a^ (°C)	(*T*_d10_) ^b^ (°C)	(*T*_d50_) ^b^ (°C)	(*T*_p_) ^c^ (°C)	(*T*_m_) ^d^ (°C)	∆*E*_c_ (kJ/mol)
PVDF	451	476	446	471	138.1	169.2	−303
HDPE	422	461	376	411	118.2	135.0	−153
F7E3	447	480	421	464	138.2	167.7	−326
F7E3-1	457	481	427	475	140.6	167.9	−330
F7E3-2	458	484	435	476	140.1	168.4	−363
F7E3-3	463	491	449	489	140.7	168.5	−374

^a^ in N_2_; ^b^ in air; ^c^ 5 °C/min cooling; ^d^ 5 °C/min pre-cooled.

**Table 4 nanomaterials-09-00361-t004:** Mechanical properties and electrical resistivity of the selected samples.

Samples	Properties			
	Young’s Modulus (σ) (MPa)	Flexural Modulus (σ) (MPa)	Impact Strength (σ) (J/m)	Electrical Resistivity (Ω-cm)
PVDF	1216 (90)	927 (28)	17.9 (2.10)	1.03 × 10^11^
HDPE	665 (41)	744 (36)	10.5 (0.98)	3.87 × 10^11^
F7E3	575 (66)	815 (29)	6.4 (0.94)	1.95 × 10^11^
F7E3-1	672 (34)	921 (18)	7.8 (0.73)	3.85 × 10^10^
F7E3-2	697 (29)	1010 (25)	9.9 (0.45)	2.32 × 10^9^
F7E3-3	654 (41)	937 (34)	8.5 (0.52)	1.79 × 10^8^

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
