# Peer review of "Graphene Nanoplatelet-Reinforced Poly(vinylidene fluoride)/High Density Polyethylene Blend-Based Nanocomposites with Enhanced Thermal and Electrical Properties"

_nanomaterials, 2019, doi:10.3390/nano9030361_

Round 1
Reviewer 1 Report
In this work, authors prepared Graphene Nanoplatelet-PVDF/HDPE nanocomposites. Morphology, Thermal, electrical and structural properties of the hybrid composites were characterized.
The manuscript has some interesting arguments but still need major revision before publication.
C1. Please elaborate “TG+TG-“ and “all-trans(TTTT)” terms in line 51.
C2. The major concerned about this paper is its novelty. Please explain carefully why the combination of PVDF and HDPE is better than neat PVDF. Table 4 shows the better mechanical properties of PVDF. Thermal properties of PVDF are not so different even with the addition of GNP.
C3. Distribution of GNP is shown by arrows in SEM image, which is not convincing. Please provide TEM images of GNP as the proof of good dispersion into the composite.
C4. Authors mentioned about the radical scavenging behavior of GNP but do not provide any reason, why this enhanced the thermal properties? Line 191
C5. In nanocomposites, the volume resistivity is of main interest compared to surface resistivity. Please provide the volume resistivity of the samples in table 4.
C6. For mechanical testing (tensile and flexural), what was the specimen size?
Author Response
Point 1: Please elaborate “TG+TG-“ and “all-trans(TTTT)” terms in line 51.
Response 1: Thanks for the valuable suggestion. As suggested by the reviewer, the following explanation is given in the revised manuscript (page 2, line 51 - 53): The non-polar α-form with trans-gauche-trans-gauche (TG+TG-) linkage conformation is the most stable and encountered polymorph. The polar β-form crystal with all-trans (TTTT) zig-zag conformation is the most attractive polymorph because of its unique piezo- and pyroelectric properties, which allow PVDF applications in sensor and actuators.
Point 2: The major concerned about this paper is its novelty. Please explain carefully why the combination of PVDF and HDPE is better than neat PVDF. Table 4 shows the better mechanical properties of PVDF. Thermal properties of PVDF are not so different even with the addition of GNP.
Response 2: Thanks for the comments. PVDF/HDPE blends are more suitable than the neat PVDF for lithium-ion battery applications. The PVDF/HDPE blend membranes show better rate capability and higher discharge capacity than that of neat PVDF membrane. In addition, PVDF/HDPE blends are cost effective compared to the neat PVDF. The mechanical properties of the blend are inferior to the neat PVDF due to the immiscible character, but slightly increase with increasing GNP content as compared to the blend. Thermal properties are slightly improved with the addition of GNP in the blend.
PVDF/HDPE blends for lithium-ion application were reported in the following paper.
Liu, H.; Liu, J.; Wang, J. Novel polymer electrolyte based on PVDF/HDPE blending for lithium-ion battery. Mater. Lett. 2013, 99, 164-167.
Point 3: Distribution of GNP is shown by arrows in SEM image, which is not convincing. Please provide TEM images of GNP as the proof of good dispersion into the composite.
Response 3: Thanks for the suggestion. The dimensions of GNP used in this study are large enough, and thus it is easy to observe the well dispersed GNP within the polymer matrix by SEM. In fact, it is kind of difficult to show the dispersion of GNP within the composite by TEM. (page 4, line 161 - 162). The newly included SEM image (inset of Fig. 1(b)) reveal clearly the dispersion of GNP in the PVDF matrix.
Point 4: Authors mentioned about the radical scavenging behavior of GNP but do not provide any reason, why this enhanced the thermal properties? Line 191
Response 4: The thermal stability increased for the GNP-added composites was attributed to the high aspect ratio of GNP which served as a barrier and then prevented the emission of gaseous molecules during the thermal degradation. (page 4, line 194-197) In addition, the radical scavenging function of GNP could inhibit the degradation process of the organic polymers.
Point 5: In nanocomposites, the volume resistivity is of main interest compared to surface resistivity. Please provide the volume resistivity of the samples in table 4.
Response 5: As requested by the reviewer, the volume resistivity values of the samples are given in the revised manuscript.
Point 6: For mechanical testing (tensile and flexural), what was the specimen size?
Response 6: For the tensile and flexural tests, the specimen size was followed ASTM D638 (dumbbell shape thickness: 2 mm) and ASTM D790 (rectangular shape thickness: 1.7 mm) standard, respectively. (page 3, line 140)

Reviewer 2 Report
The manuscript entitled “Graphene nanoplatelet-reinforced poly(vinylidene fluoride)/high density polyethylene blend-based nanocomposites with enhanced thermal and electrical properties” is an interesting study about polymer nanocomposites designed for electromechanical and capacitors applications. The manuscript is very clear and well described. In my opinion, this work is very complete, but It needs some revision. Following are some comments about the manuscript:
1. Materials and methods:
In the introduction, authors explain that the GNP dispersion is one of the key factors to achieve nanocomposites with advanced performance. However, they do not explain in the manuscript how these GNP were dispersed and the dispersant used during the mixing of both polymers and the filler. Please, specify this part.
Some references about this subject are:
· González-García CM, González-Martín ML, Gómez-Serrano V, Bruque JM, Labajos-Broncano L. Determination of the free energy of adsorption on carbon blacks of a nonionic surfactant from aqueous solutions. Langmuir 2000;16:3950–6.
· E. Enríquez, J.F. Fernández, M.A. de la Rubia. Highly conductive coatings of carbon black/silica composites obtained by a sol–gel process. Carbon 50(2012) 4409-4417 https://doi.org/10.1016/j.carbon.2012.05.019
2. Phase morphology and selective localization of GNP:
Authors found that the domain size of HDPE decreases after the addition of GNP, but why this decrease occurs?
In figure 1, the scale in the SEM images is not well observed. Please, improve these images.
3. Crystal structure, FTIR and Raman spectra:
Authors explain that the addition of GNP does not affect to the PVDF and HDPE crystallization, however, it is observed a change in the intensities ratio (mainly between (020) and (110)) in PVDF peaks in the blends. What these changes mean?
4. Mechanical properties:
Authors give electrical resistivity values of blends in table 4, but in my opinion, they should give also some values of similar polymers of literature, in order to compare.
For this reason, I think this manuscript needs minor revisions.
Author Response
Point 1: In the introduction, authors explain that the GNP dispersion is one of the key factors to achieve nanocomposites with advanced performance. However, they do not explain in the manuscript how these GNP were dispersed and the dispersant used during the mixing of both polymers and the filler. Please, specify this part.
Response 1: Thanks for the valuable comment. No dispersant was used in this study. Thermo Haake PolyDrive mixer (R600) (counter rotating rotors) was used for the preparation of samples. Many papers already reported good results by using melt-mixing method at proper conditions and the GNP was dispersed well.
Point 2: Authors found that the domain size of HDPE decreases after the addition of GNP, but why this decrease occurs? In figure 1, the scale in the SEM images is not well observed. Please, improve these images.
Response 2: The added-GNP might have prevented the aggregation of HDPE domains forming large HDPE domains. (page 4, line 156-157)
Similar results were reported in the following cited paper [Ref. 3].
Sivanjineyulu, V.; Behera, K.; Chang Y.H.; Chiu, F.C. Selective localization of carbon nanotube and organoclay in biodegradable poly(butylene succinate)/polylactide blend-based nanocomposites with enhanced rigidity, toughness and electrical conductivity. Compos. Part A. Appl. Sci. Manuf. 2018, 114, 30-39.
Authors have improved the quality of SEM images accordingly. See in revised manuscript.
Point 3: Authors explain that the addition of GNP does not affect to the PVDF and HDPE crystallization, however, it is observed a change in the intensities ratio (mainly between (020) and (110)) in PVDF peaks in the blends. What these changes mean?
Response 3: The change in intensity ratio [(mainly between (020) and (110)] of PVDF in the blend/composites is because the growth of (020) plane was somehow retarded compared to the growth of (110) plane for PVDF crystals. (page 13, line 304-306)
Point 4: Authors give electrical resistivity values of blends in table 4, but in my opinion, they should give also some values of similar polymers of literatures, in order to compare.
Response 4: Thanks for the suggestion. PVDF shows the electrical resistivity value similar to the below-published papers, as already addressed in the revised manuscript [Ref. 31].
1. Chiu, F.C.; Chen, Y.J. Evaluation of thermal, mechanical, and electrical properties of PVDF/GNP binary and PVDF/PMMA/GNP ternary nanocomposites. Compos. Part A: Appl. Sci. Manuf. 2015, 68, 62-71.

Reviewer 3 Report
The manuscript “Graphene Nanoplatelet-Reinforced Poly(vinylidene fluoride)/High Density Polyethylene Blend-Based Nanocomposites with Enhanced Thermal and Electrical Properties” investigates thermal, electrical and mechanical properties of PVDF/HDPE blends and PVDF/HDPE/GNP composites. The manuscript is well written and results described clearly with good supporting arguments and references. I recommend the manuscript can be accepted for publications in “Nanomaterials” after a few minor corrections:
1) Scale bar in figure 1 is not visible, please improve this.
2) Page 11, line 284: typing error F7E3-P1, I think it should be F7E3-1.
3) General comment: I think the manuscript can be improved further if authors can use the prepared composites for some application for eg. micro-electromechanical devices or storage capacitors?
Author Response
Point 1: Scale bar in figure 1 is not visible, please improve this.
Response 1: Authors have improved the quality of SEM images accordingly.
Below is the newly supplied images:
See in revised manuscript.
Point 2: Page 11, line 284: tying error F7E3-P1, I think it should be F7E3-1.
Response 2: The F7E3-P1 was replaced by the F7E3-1 in the text at appropriate place accordingly. (page 11, line 289)
Point 3: General comment: I think the manuscript can be improved further if authors can use the prepared composites for some application for eg. micro-electromechanical devices or storage capacitors?
Response 3: Thanks for the valuable suggestion. Authors could not supply data for the micro-electromechanical devices or storage capacitors application for the prepared composites at this moment. But in the future studies, the suggestion will be considered.

Round 2
Reviewer 1 Report
Authors revised the manuscript according to the comments from reviewers, therefore it is recommended to publish in its present form.
Author Response
Thanks.